

# Machine learning analysis to identify the association between risk factors and onset of nosocomial diarrhea: a retrospective cohort study

Ken Kurisu[1,2], Kazuhiro Yoshiuchi[1], Kei Ogino[1,2] and Toshimi Oda[2]

[1] Department of Stress Sciences and Psychosomatic Medicine, Graduate School of Medicine, The University of Tokyo, Tokyo, Japan
[2] Department of Infectious Diseases, Showa General Hospital, Tokyo, Japan

## ABSTRACT

**Background**. Although several risk factors for nosocomial diarrhea have been identified, the detail of association between these factors and onset of nosocomial diarrhea, such as degree of importance or temporal pattern of influence, remains unclear. We aimed to determine the association between risk factors and onset of nosocomial diarrhea using machine learning algorithms.

**Methods**. We retrospectively collected data of patients with acute cerebral infarction. Seven variables, including age, sex, modified Rankin Scale (mRS) score, and number of days of antibiotics, tube feeding, proton pump inhibitors, and histamine 2-receptor antagonist use, were used in the analysis. We split the data into a training dataset and independant test dataset. Based on the training dataset, we developed a random forest, support vector machine (SVM), and radial basis function (RBF) network model. By calculating an area under the curve (AUC) of the receiver operating characteristic curve using 5-fold cross-validation, we performed feature selection and hyperparameter optimization in each model. According to their final performances, we selected the optimal model and also validated it in the independent test dataset. Based on the selected model, we visualized the variable importance and the association between each variable and the outcome using partial dependence plots.

**Results**. Two-hundred and eighteen patients were included. In the cross-validation within the training dataset, the random forest model achieved an AUC of 0.944, which was higher than in the SVM and RBF network models. The random forest model also achieved an AUC of 0.832 in the independent test dataset. Tube feeding use days, mRS score, antibiotic use days, age and sex were strongly associated with the onset of nosocomial diarrhea, in this order. Tube feeding use had an inverse $U$-shaped association with the outcome. The mRS score and age had a convex downward and increasing association, while antibiotic use had a convex upward association with the outcome.

**Conclusion**. We revealed the degree of importance and temporal pattern of the influence of several risk factors for nosocomial diarrhea, which could help clinicians manage nosocomial diarrhea.

Corresponding author
Kazuhiro Yoshiuchi, kyoshiuc-tky@umin.ac.jp

## INTRODUCTION

Nosocomial diarrhea is a common problem among hospitalized patients. It increases length of stay and healthcare costs (*Kyne et al., 2002a*) and is important for hospital infection control. Several risk factors for nosocomial diarrhea have been identified, including history of hospitalization, gastrointestinal surgery, severity of disease, and tube feeding, proton pump inhibitor (PPI), histamine 2-receptor antagonist (H2RA), and antibiotic use (*McFarland, 1995*; *Kyne et al., 2002b*; *Thorson, Bliss & Savik, 2008*; *Arevalo-Manso et al., 2014*; *Eze et al., 2017*; *Thabit, Varugehese & Levine, 2019*).

However, details about association between these risk factors and the onset of nosocomial diarrhea remains unclear. There are few reports about the association between the risk of onset of nosocomial diarrhea and the duration of antibiotic, PPI, H2RA, and tube feeding use. Clinical knowledge of relative importance or temporal pattern of the influence of risk factors for nosocomial diarrhea could result in minimizing administration of these drugs or considering administration of probiotics for preventing nosocomial diarrhea (*Hempel et al., 2012*; *Goldenberg et al., 2017*).

Recently, there have been many reports using machine learning algorithms in the medical field, such as random forest, support vector machine (SVM), or radial basis function (RBF) network (*Bair et al., 2013*; *DuBrava et al., 2017*; *Kimura et al., 2019*; *Halladay, Sillner & Rudolph, 2018*; *Le, Ho & Ou, 2018*; *Le et al., 2019a*; *Tamune et al., 2019*; *Cho et al., 2018*; *Ding et al., 2018*; *Zarbakhsh & Addeh, 2018*). These models are non-linear and non-monotonous. They can deal with variables that have a complex association, such as a *U*-shaped or convex association, with the outcome. They can analyze the duration of exposure and presence or absence of exposure simultaneously. Traditional statistical models, such as a logistic regression or Cox regression model, do not have such properties. In addition, some machine learning models have high interpretability because they can visualize the variable importance (*Breiman, 2001*; *Fisher, Rudin & Dominici, 2018*) or association between the variables and the outcome by partial dependence plots (*Friedman, 2001*).

We hypothesized that such machine learning algorithms could reveal the unknown association, such as degree of importance and temporal pattern of the influence of risk factors for nosocomial diarrhea, which would be helpful for clinicians. The present study aimed to determine the association between the risk factors and onset of nosocomial diarrhea using machine learning algorithms. We used the data of patients hospitalized with acute cerebral infarction because they often have several risk factors for nosocomial diarrhea, such as PPI, H2RA, and tube feeding use.

## MATERIALS & METHODS

### Ethics approval

This study was approved by the Institutional Review Board of Showa General Hospital (approval number: REC-180). In the present study, because of the anonymous nature of the data and the non-invasive study, the requirement for informed consent was waived. Instead, we released the research project on the website of Showa General Hospital so that patients could reject utilization of their data.

### Design and study population

This retrospective cohort study was conducted in Showa General Hospital, a single tertiary center in Japan. We collected data from electronic medical records.

We included patients admitted from April 2017 to March 2018 for acute cerebral infarction, except for those diagnosed with subtype 4 infarction according to the Trial of Org 10172 in Acute Stroke Treatment (TOAST) classification (*Adams Jr et al., 1993*), who stayed for more than 3 days. To our knowledge, there is no theoretical method to determine the sample size in machine learning models. Therefore, we included patients admitted during one fiscal year. Cerebral infarction of subtype 4 according to the TOAST classification includes diseases, such as vertebral artery dissection. Because cases of subtype 4 cerebral infarction are relatively rare (*Kolominsky-Rabas et al., 2001*), we did not use them to reduce patient heterogeneity. The present study focused on nosocomial diarrhea, which occurs more than 3 days after admission (*Bauer et al., 2001*). Therefore, patients who stayed for more than 3 days were eligible.

The exclusion criteria were history of abdominal surgery, gastrointestinal disease as a comorbidity, regular use of laxatives, and the onset of diarrhea within 3 days from admission. Patients with a history of abdominal surgery and those with gastrointestinal disease as a comorbidity were excluded to streamline the analysis. Patients who were using laxatives regularly were excluded because it was difficult to conclude whether diarrhea in these patients was caused by the regular use of laxatives. Patients who presented with diarrhea within 3 days from admission were excluded for the same reason as those who left the hospital within 3 days from admission.

### Outcome

According to the criteria established by the *WHO (2018)* and used in previous research (*Arevalo-Manso et al., 2014*), diarrhea was defined as the passage of 3 or more liquid stools or stools of types 5–7 according to the Bristol Stool Form Scale (*Lewis & Heaton, 1997*) within 24 h. If patients had diarrhea but were using laxatives temporarily within 3 days of the onset of diarrhea, we considered that their diarrhea was likely to be caused by the laxatives, regardless of the risk factors, and we categorized them into the non-diarrhea group.

The observational period was from admission to the onset of diarrhea in patients who had diarrhea and from admission to discharge in patients who did not have diarrhea.

We used C Diff Quik Chek (Abbott, Lake Bluff, Ill.), a tool of enzyme immunoassay for *Clostridioides difficile* (CD) glutamate dehydrogenase (GDH) antigen and CD toxins A and

B, as the diagnostic tool for CD infection (CDI). It is reported that sensitivity and specificity for GDH antigen is over 90%, sensitivity for CD toxin is almost 50%, and specificity for CD toxin is over 90% in Japanese hospitals (*Kawada et al., 2011*; *Kosai et al., 2017*; *Morinaga et al., 2018*).

## Variables

The following seven variables were considered as risk factors or confounding factors for nosocomial diarrhea and were included in the analysis: age, sex, severity of illness, and number of days of use of tube feeding, PPI, H2RA, and antibiotics (*McFarland, 1995*; *Kyne et al., 2002b*; *Thorson, Bliss & Savik, 2008*; *Arevalo-Manso et al., 2014*; *Eze et al., 2017*; *Thabit, Varugehese & Levine, 2019*). The modified Rankin Scale (mRS) (*Van Swieten et al., 1988*) was used as the index of disease severity and was scored 2 days after admission. The number of days of exposure to the risk factors was calculated at the end of the observational period: at the onset of diarrhea for patients in the diarrhea group and at discharge for patients in the non-diarrhea group. For patients without exposure, each variable was set to 0.

## Analysis of demographic data

All analyses were conducted using the open source software R version 3.6.1 (*R Core Team, 2019*). The threshold for statistical significance was set to $P < .05$.

We used the Student's $t$ test to compare the averages of continuous variables and the chi-squared test to compare the proportions of categorical variables between the diarrhea and non-diarrhea groups. The multicollinearity of the independent variables was evaluated using variance inflation factors (VIFs).

## Model selection

We developed machine learning models that classified the data into diarrhea group or non-diarrhea group. First, we split the data for 9 months (from April 2017 to December 2017) into the training dataset, and data for 3 months (from January 2018 to March 2018) into the test dataset. Next, based on the training dataset, we developed the random forest, SVM, and RBF network model, using R libraries named "randomForest" (version 4.6.14), "e1071" (version 1.7.2), and "RSNNS" (version 0.4.11), respectively.

By calculating the area under the curve (AUC) of the receiver operating characteristic curve using 5-fold cross-validation (*DuBrava et al., 2017*; *Le et al., 2019b*; *Le, Ho & Ou, 2017*; *Ukita, Yoshida & Ohki, 2019*), we performed feature selection and hyperparameters optimization to determine the models. Calculation of AUC was performed using an R library named 'ROCR' (version 1.0.7). The sets of hyperparameters were determined so that the AUC calculated by the cross-validation was optimized. Feature selection was performed using the backward elimination method for the variables filtered by statistical significance between the diarrhea group and non-diarrhea group within the training dataset (*Tangaro et al., 2015*; *Milošević et al., 2017*).

According to the final AUC of each model, we selected the optimal model for the following analysis. The selected model was also validated using the independent test dataset.

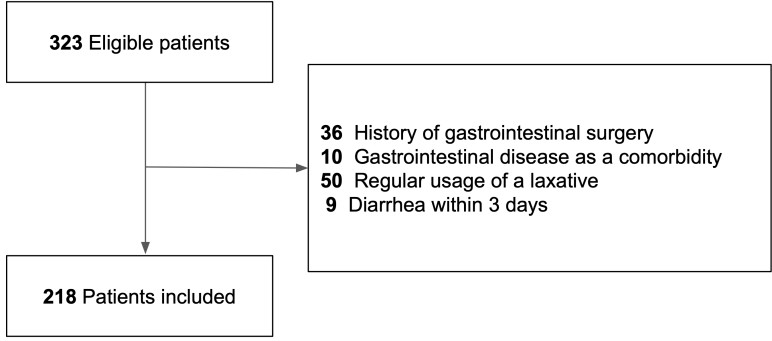

**Figure 1 Flow chart of the study cohort.** Flow chart shows the number of included and excluded patients and the reasons for exclusion.

## Visualization of variables' nature

We visualized the variable importance in the selected model. We also visualized the association between the variables and the outcome by partial dependence plots using an R library named "pdp" (version 0.7.0).

## RESULTS

### Participant characteristics

Three hundred twenty-three patients with acute cerebral infarction were potentially eligible for inclusion in the present study, and after applying the exclusion criteria, 218 patients were included (Fig. 1).

Table 1 shows the descriptive data for the diarrhea and non-diarrhea groups. Among the 218 patients, 48 had diarrhea during the observation period. In the diarrhea group, the test of CDI was performed in 12 (25%) patients. Among them, two (4%) patients were positive for the GDH antigen and none were positive for the CD toxin. The time of onset of nosocomial diarrhea ranged from 4 to 45 days from admission. There were no missing data for each variable. The patients in the diarrhea group were older, had a higher mRS score, had higher rates of antibiotic, tube feeding, and H2RA use. They also had longer duration of antibiotic and tube feeding use than those in the non-diarrhea group.

The VIF for each variable was <2, indicating there was no multicollinearity.

### Model selection

Comparison of the model performance is shown in Table 2. Three variables (sex, H2RA use days, and PPI use days) were not significant between the diarrhea group and non-diarrhea group within the training dataset, which is similar to the results of the whole dataset shown in Table 1. Because eliminating some variables improved the performance, these variables were not used in the following analysis. The optimal hyperparameters were also determined for each model and used in the following analysis.

The three models achieved almost the same AUC on the 5-fold cross-validation. The random forest model achieved a higher performance than the other two models. In addition, visualization of variable importance and partial dependence plots are widely used

**Table 1  Characteristics of the study population.**

| Variable | Diarrhea group (n = 48) | Non-diarrhea group (n = 170) | P value |
|---|---|---|---|
| Observational period, median (range), days | 13 (4–45) | 17.5 (5–59) | |
| Age, mean (SD), years | 80.6 (10.0) | 75.6 (10.9) | **.004**[a] |
| mRS score, mean (SD) | 4.25 (0.86) | 2.45 (1.50) | **<.001**[a] |
| Male sex, n (%) | 35 (73) | 97 (57) | .047[b] |
| Antibiotic use, n (%) | 25 (52) | 19 (11) | **<.001**[b] |
| Number of days of use, median (range) | 3 (0–18) | 0 (0–14) | **<.001**[a] |
| Tube feeding use, n (%) | 25 (52) | 6 (4) | **<.001**[b] |
| Number of days of use, median (range) | 1 (0–33) | 0 (0–42) | **<.001**[a] |
| PPI use, n (%) | 33 (69) | 120 (71) | .81[b] |
| Number of days of use, median (range) | 6 (0–45) | 11 (0–59) | .09[a] |
| H2RA use, n (%) | 31 (65) | 77 (45) | **.02**[b] |
| Number of days of use, median (range) | 2 (0–32) | 0 (0–32) | .36[a] |
| Tests for *Clostridioides difficile* (CD) infection | | | |
| Examination conducted, n (%) | 12 (25) | | |
| CD toxin A and B positive, n (%) | 0 (0) | Not applicable | |
| GDH antigen positive, n (%) | 2 (4) | | |
| No examination conducted, n (%) | 36 (75) | | |

Notes.

H2RA, histamine 2-receptor antagonist; mRS, modified Rankin Scale; PPI, proton pump inhibitor; GDH, glutamate dehydrogenase.

[a]Student's *t* test.

[b]Chi-squared test.

**Table 2  The performance of the machine learning models.**

| | AUC | Variables eliminated | Hyperparameters |
|---|---|---|---|
| 5-fold cross-validation | | | |
| Random forest | 0.944 | PPI, H2RA | Number of features: 1 |
| | | | Number of trees: 500 |
| SVM | 0.937 | H2RA | Gamma: 0.0063 |
| | | | Cost: 0.016 |
| RBF network | 0.934 | Sex | Size of hidden layer: 22 |
| Independent test dataset | | | |
| Random forest | 0.832 | PPI, H2RA | Number of features: 1 |

Notes.

SVM, Support vector machine; RBF, Radial basis function; AUC, Area under the curve.

in the random forest model (*Bair et al., 2013*; *DuBrava et al., 2017*; *Kimura et al., 2019*; *Halladay, Sillner & Rudolph, 2018*), unlike SVM or RBF network. Therefore, among the 3 models, we selected the random forest model for the following analysis.

The random forest model also achieved an AUC of 0.83 in the dependent test dataset, which was suitable for the following discussion regarding each variable.

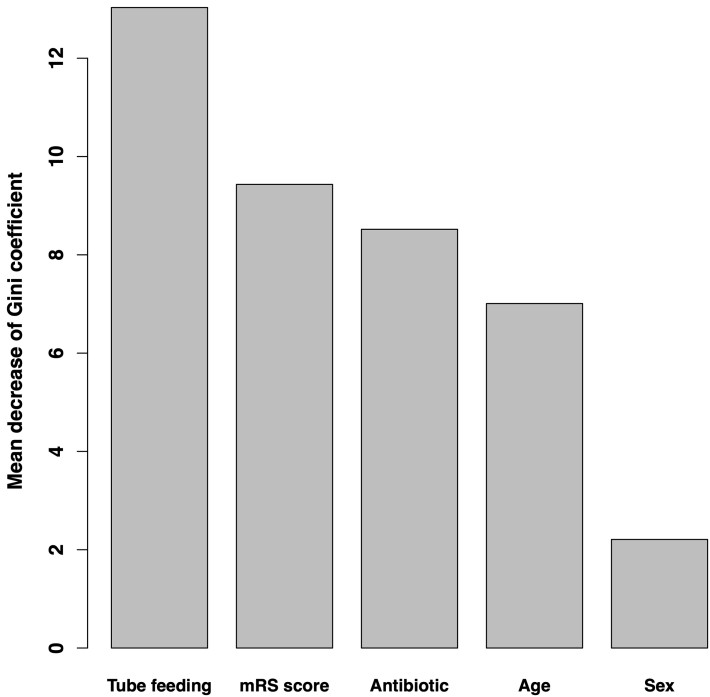

**Figure 2** **Variable importance according to mean decrease in Gini coefficient.** Bar graphs show the mean decrease in the Gini coefficient of each variable, which is considered as the index of importance. mRS, modified Rankin Scale.

## Variable importance

Variable importance according to the mean decrease of Gini coefficient is shown in Fig. 2. The order of importance was as follows: tube feeding use days, mRS score, antibiotic use days, age, and sex.

## Partial dependence plots

Partial dependence plots of each variable are shown in Fig. 3. The value of the $y$-axis in the Figure was calculated using the partial dependence function, which approximately represented the probability of onset of nosocomial diarrhea.

Tube feeding use had an almost inverse $U$-shaped association with the outcome; the use of a feeding tube drastically increased the risk of nosocomial diarrhea in the first few days, but gradually decreased it thereafter. Overall, patients without use of tube feeding had a lower risk than those with such use. The mRS score increased the risk of nosocomial diarrhea with a convex downward and increasing association; especially patients with an mRS score >3 had a high risk of nosocomial diarrhea. Antibiotic use had convex upward association; the use of antibiotics rapidly increased the risk in the first few days, and slowly increased it thereafter. Patients without use of antibiotics had a lower risk than those with such use. The association between age and nosocomial diarrhea was convex downward and increasing; especially patients aged >90 years had a high risk of nosocomial diarrhea. Male sex slightly increased the risk of nosocomial diarrhea.
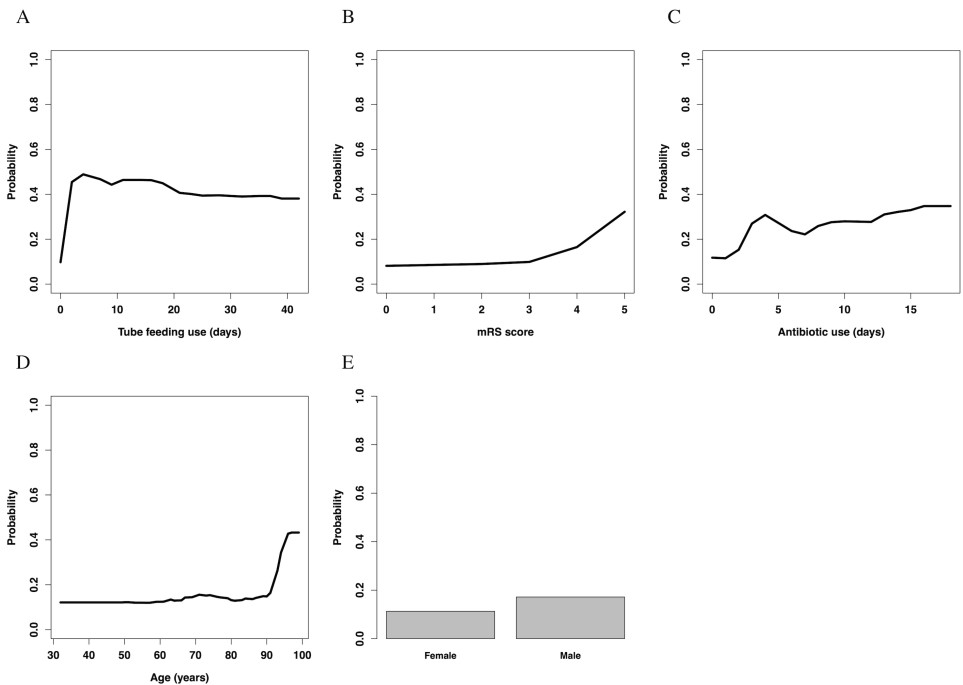

**Figure 3** **Partial dependence plots.** (A) Temporal changes in the influence of tube feeding use. (B) The association between mRS score and influence. (C) Temporal changes in the influence of antibiotics use. (D) The association between age and influence. (E) The association between sex and influence. mRS, modified Rankin Scale.

## DISCUSSION

In our study, the random forest model achieved high performance. The model showed that tube feeding use days, mRS score, antibiotic use days, age, and sex were important, in this order. Tube feeding use had an inverse $U$-shaped association with the outcome. The mRS score and age showed a convex downward and increasing association. Antibiotic use showed a convex upward association with the outcome. Male patients had a slightly higher risk of nosocomial diarrhea.

Tube feeding had the strongest association with the onset of nosocomial diarrhea in the analysis. This result is consistent with that in previous studies (*McFarland, 1995*; *Thorson, Bliss & Savik, 2008*; *Arevalo-Manso et al., 2014*). However, the model showed that when it was used for >4 days, the risk conversely decreased, which could be a new finding. These results implied that if nosocomial diarrhea occurred with prolonged use of tube feeding, clinicians should consider a differential diagnosis other than diarrhea owing to tube feeding.

The mRS score, which was used for disease severity, was the second strongest factor associated with nosocomial diarrhea. Patients with a low mRS score were often transferred on foot or with light assistance, whereas those with a high mRS score were often transferred by a wheelchair or on a stretcher. This might imply that activities of daily living, such as mode of movement, were associated with the onset of nosocomial diarrhea. Previous

research revealed that regular exercise prevents episodes of diarrhea (*Ma et al., 2014*), and another study revealed that exercise alters the composition and functional capacity of gut microbiota (*Mailing et al., 2019*). In addition, the severity of disease was reported to have an association with nosocomial diarrhea in previous studies (*Kyne et al., 2002b*; *Thorson, Bliss & Savik, 2008*). These results may provide a mechanism for the present findings.

Antibiotic use had an almost convex upward association with the outcome. The use of antibiotics rapidly increased the risk of nosocomial diarrhea in the first 4 days. After 5 days of use, its use slowly increased the risk. Although previous studies revealed that antibiotic use is an important risk factor for nosocomial diarrhea (*McFarland, 1995*; *Arevalo-Manso et al., 2014*; *Eze et al., 2017*), this convex upward association could be a new finding. A recent study showed that the diversity of gut microbiome is affected by antibiotic administration and that the component of gut microbiome changes with time (*Bulow et al., 2018*). The former rapid slope and latter gradual slope in the present study might arise owing to the effect of different microorganisms. Another recent study showed that the median of onset of CDI was about one week after antibiotic therapy (*Thabit, Varugehese & Levine, 2019*), which does not conflict with our result, although our study had a limitation regarding CDI. Because our study might include a small number of patients with CDI, as discussed below, these results might apply mainly to antibiotic-associated nosocomial diarrhea other than CDI. After all, the result of the present study might imply that clinicians should consider discontinuing antibiotics as soon as possible to prevent nosocomial diarrhea.

The association between age and nosocomial diarrhea was convex downward and increasing. Especially, age >90 years rapidly increased the risk of nosocomial diarrhea. This result is consistent with that of a previous study (*McFarland, 1995*).

Male patients had a slightly higher risk of nosocomial diarrhea than female patients. However, to our knowledge, there is no rational explanation of this result; it might imply that the collected data had some bias, and that the sex variable acted like a confounding factor.

Number of days of PPI and H2RA use were not significantly different between the diarrhea group and the non-diarrhea group. They were also removed from the analysis by backward elimination, which implied that these variables were not important for prediction. Although previous studies showed that PPI and H2RA use were independent risk factors for nosocomial diarrhea (*Eze et al., 2017*), their importance was relatively low in the present study. The insignificance of antacid drugs was also observed in another study focusing on CDI (*Thabit, Varugehese & Levine, 2019*). These results might imply that the effect of acid suppression therapy on risk of nosocomial diarrhea is lower than those of other risk factors, such as tube feeding, disease severity and antibiotics.

The primary limitation is that the causes of diarrhea were not precisely diagnosed and not included in the analysis. In particular, the present study included only 2 GDH antigen positive patients and no patients who were positive for CD toxin. Even considering the low sensitivity of our diagnosis tool for CDI (*Kawada et al., 2011*; *Kosai et al., 2017*; *Morinaga et al., 2018*), this rate is lower than that reported in hospitals in the US, where there were 20–30% cases of nosocomial antibiotic-associated diarrhea (*McDonald et al., 2018*). We considered that there are two rational explanations for this result. The first is that there

were many patients with CDI overlooked in the study. A systematic review shows that prevalence of CDI in Japanese hospitals is lower than that of US and European countries because of under-diagnosis (*Riley & Kimura, 2018*). Another study shows that numerous patients with CDI are being overlooked due to inadequate diagnostic testing in Japan (*Kato et al., 2019*). These studies could support the first explanation. Another explanation is that there was actually a small number of patients with CDI. As reported in a previous study (*Morii et al., 2018*), antimicrobial stewardship was widely implemented in Showa General Hospital from October 2010. Because antimicrobial stewardship greatly reduces the frequency of CDI (*Baur et al., 2017*), the number of patients with CDI might actually be low, as reported in Table 1. However, these hypotheses could not be verified retrospectively, and we consider this point to be the primary limitation. Other limitations are as follows: the present study was conducted in a single center, so external validity was not confirmed; the types and amounts of medications and tube feeding were not considered; and some risk factors or confounding factors, such as serum albumin level, were not considered in the analysis.

## CONCLUSIONS

We revealed the degree of importance and temporal pattern of the influence of several risk factors for nosocomial diarrhea, such as tube feeding, mRS score, antibiotic use, and age. These findings could help clinicians manage nosocomial diarrhea.

## ACKNOWLEDGEMENTS

We thank Drs. Eri Fukao and Yutaka Honma for their helpful advice on the methods of neurological assessments. We also thank Dr. Jumpei Ukita for his helpful advice on the machine learning analysis.

### Funding
The authors received no funding for this work.

### Competing Interests
The authors declare there are no competing interests.

### Author Contributions
- Ken Kurisu conceived and designed the experiments, performed the experiments, analyzed the data, contributed reagents/materials/analysis tools, prepared figures and/or tables, authored or reviewed drafts of the paper, approved the final draft.
- Kazuhiro Yoshiuchi conceived and designed the experiments, analyzed the data, contributed reagents/materials/analysis tools, authored or reviewed drafts of the paper, approved the final draft.

- Kei Ogino and Toshimi Oda conceived and designed the experiments, contributed reagents/materials/analysis tools, authored or reviewed drafts of the paper, approved the final draft.

## Human Ethics

The following information was supplied relating to ethical approvals (i.e., approving body and any reference numbers):

This study was approved by the Institutional Review Board of Showa General Hospital (approval number: REC-180).

## Data Availability

The raw data and code are available in the Supplemental Files.

## Supplemental Information

Supplemental information for this article can be found online at http://dx.doi.org/10.7717/peerj.7969#supplemental-information.

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
