# Peer review of "Machine learning analysis to identify the association between risk factors and onset of nosocomial diarrhea: a retrospective cohort study"

_PeerJ, doi:10.7717/peerj.7969_

## Round 0.1 · original submission · Major Revisions

I hope you can easily revise your interesting manuscript. Please let me know if you have any problem in doing so.

·

Basic reporting

Well-structured paper. But I have some comments on the discussion and table 1 that I listed under the comments below.

Experimental design

No comment.

Validity of the findings

Findings seem valid and interesting; however, I have one concern (comment no. 4) and a suggestion to add an important finding regarding Clostridioides (formerly Clostridium) difficile infection (comment no. 5). Please find the details in my comments below.

Additional comments

This is an interesting and well-designed study by Kurisu and colleagues assessing risk factors associated with nosocomial diarrhea in patients admitted with cerebral infarction. However, I have some comments and questions:

Major comments:
1. On Table 1: Regarding the variables of antibiotic use, Tube feeding use, PPI, and H2RA use, what do the P values represent? Do they represent the difference between the two groups in terms of using these variables or the number of days of using them? The P values appear to be in the middle between the use of the variable and the number of days, hence, it isn't clear to which of the two it belongs. Please clarify, and if there's another P value for the other part of the variable, please add it. For example, if the P value represents the antibiotic use, then please add the P value for the difference in the number of days of antibiotic use.
2. Also on Table 1: the P value corresponding to the male sex is 0.047 which rounds to 0.05. In the medical literature, such P value (that can be rounded to 0.05) is considered insignificant. As such, I would suggest removing the part that says "included more men" on line 141 which can indicate that male sex is significantly associated with nosocomial diarrhea.
3. I wonder why the authors only examined the number of days of the use of the variables but not their use or lack of thereof. I suggest, if possible, including the presence or absence of these variables in the random forest model and then present the data of the number of days as a subgroup analysis in each group of patients where only the variables of statistical significance were present (i.e., only the variables that were significantly associated with nosocomial diarrhea are evaluated for the association of the number of days of use). Alternatively, I suggest that the authors justify and explain the rationale behind the inclusion of only the number of days in the last part of the introduction where they list the study's objectives.
4. I found it interesting that there seem to be some sort of inconsistency between the findings of Table 1 and Figure 2 that I suggest that the authors comment on in the discussion section or perhaps double check their results and statistics. The discrepancies are in the level of importance of each variable (or how strongly each is associated) with nosocomial diarrhea. For instance, PPI use (or the number of days; please address my comment no. 1) has a P value of 0.81 on Table 1 while H2RA use has a significant P value of 0.02. On the other hand, Figure 2 shows the number of days of PPI use is more associated with nosocomial diarrhea than the number of days of H2RA use! In my opinion, if the use of PPI was actually not significantly associated with nosocomial diarrhea as presented on Table 1, then I think it might be inappropriate to include the association of the number of days of their use in the analysis since it is not a variable that's associated with the outcome in question (nosocomial diarrhea) to start with. Again, since sex is supposed to be not significantly associated with diarrhea given a P value that rounds to 0.05 per Chi-square test, I suggest removing this variable from the random forest modeling analysis and thus removing it from Figure 2.
5. I don't see anywhere in the manuscript the assessment of the presence of Clostridioides (formerly Clostridium) difficile infection in the group of patients with diarrhea since it accounts for 20-30% of cases of nosocomial diarrhea (McDonald LC, et al. Clin Infect Dis. 2018;66 (7):987-94). In fact, many published studies showed the association of the use of antibiotics, PPI, and H2RA with this infection, and these are the same variable evaluated in the present study. As such, I strongly suggest the authors to include the number of patients with C. difficile infection on Table 1 under the "Diarrhea group" column and to type "Not applicable" under the "Non-diarrhea group" column. Adding such information would aid the clinicians to consider (or not to) testing for C. difficile infection in patients with cerebral infarction who develop diarrhea. If this information is to be included, then I advise the authors to also include the type of C. difficile detection test used by the microbiology lab in their hospital (E.g., PCR or toxin immunoassay) under the methods section of the manuscript. Additionally, a comment on C. difficile infection would be highly recommended to be added to the discussion section when discussing the association of antibiotics with the infection (lines 189-198), as well as acid suppressing agents (not currently present in the discussion. See comment no. 6 below).
6. Discussion: There was no commenting on PPI and H2RA use and their association with nosocomial diarrhea, especially that H2RA use was significantly associated with diarrhea as on Table 1 and that both can be associated with C. difficile infection (a common nosocomial diarrhea as stated in comment no. 5 above) as observed in many published studies. Authors are recommended to comment on that aspect while relating their findings with data from the literature on the association of acid suppression with C. difficile infection or nosocomial diarrhea in general as they did with the other variables.
7. Discussion: The third limitation states that the causes of diarrhea were not included in the study. Although I thank the authors for disclosing this as a limitation, I would still suggest including the number of patients with C. difficile infection among the diarrhea group. Afterwards, this limitation can be either eliminated or amended to state that "the causes of diarrhea other than the presence of C. difficile infection were not included in the study" since there are other triggers for the nosocomial diarrhea other than gut infections.

Minor comments:
8. Line 175: Please correct the first sentence to say either "which was used as an indicator of diseases severity, held the second place as a factor strongly associated with nosocomial diarrhea" or "which was used as an indicator of diseases severity, was the second strong factor associated with nosocomial diarrhea."
9. Line 178: I'm not sure if "transferring" is considered a daily activity in this context. Instead, I suggest using the word "mode of movement" as it indicates the way the patient moves (or is moved) around whether with or without assistance, especially that the authors are tailing this finding with a comment on the effect of regular exercise on diarrhea.

Reviewer 2 ·

Basic reporting

- There are some language mistakes or typos that should be improved in the revised version.
- Literature references are weak. They didn't support a good research objective.
- To mention Random Forest, the authors have to cite more related published works that used it in biomedical field, such as https://doi.org/10.1002/jcc.24842 and https://doi.org/10.1016/j.ab.2018.06.011.
- Hypotheses need to be mentioned clearly.

Experimental design

- The research question is not well defined, relevant and meaningful.
- Area under the curve (AUC) and n-fold cross-validation had been used successfully in a lot of biomedical works such as https://doi.org/10.1016/j.ab.2019.02.017 and https://doi.org/10.1016/j.ab.2019.03.017. Therefore, the authors should cite more related papers to attract broader readership.
- Did the authors select the optimal parameters in their Random Forest algorithm? If selected, what was the strategy?

Validity of the findings

- Since Random Forest is one kind of machine learning technique, the authors have to explain why did they choose this algorithm. Is there any reason that helps Random Forest become a suitable algorithm for this problem? If possible, the authors should compare their performance with the state-of-the-art machine learning algorithms, i.e., SVM or RBF network.
- There is a need for validating on an external set of data.
- There is a need for discussing on some tests with p-value > 0.05 (not significant).
- The authors have to perform feature selection in their model.

Additional comments

No comment.

---

## Round 0.2 · accepted · Accept

Congraturations on your nice work !!!

·

Basic reporting

Very good

Experimental design

Very good

Validity of the findings

Very good

Additional comments

Great work revising the manuscript and precisely addressing the reviewers comments. I think it can be accepted in the current format.

Reviewer 2 ·

Basic reporting

No comment

Experimental design

No comment

Validity of the findings

No comment